# Rather a Nurse than a Physician -
# Contrastive Explanations under Investigation

**Oliver Eberle** [†∘∗]    **Ilias Chalkidis** [◇∗]    **Laura Cabello** [◇]    **Stephanie Brandl** [◇]

[†]Machine Learning Group, Technische Universität Berlin, Germany    [∘]BIFOLD, Germany
[◇]Department of Computer Science, University of Copenhagen, Denmark
oliver.eberle@tu-berlin.de, {ilias.chalkidis, lcp, brandl}@di.ku.dk

## Abstract

Contrastive explanations, where one decision is explained *in contrast to another*, are supposed to be closer to how humans explain a decision than non-contrastive explanations, where the decision is not necessarily referenced to an alternative. This claim has never been empirically validated. We analyze four English text-classification datasets (SST2, DynaSent, BIOS and DBpedia-Animals). We fine-tune and extract explanations from three different models (RoBERTa, GTP-2, and T5), each in three different sizes and apply three post-hoc explainability methods (LRP, GradientxInput, GradNorm). We furthermore collect and release human rationale annotations for a subset of 100 samples from the BIOS dataset for contrastive and non-contrastive settings. A cross-comparison between model-based rationales and human annotations, both in contrastive and non-contrastive settings, yields a high agreement between the two settings for models as well as for humans. Moreover, model-based explanations computed in both settings align equally well with human rationales. Thus, we empirically find that humans do not necessarily explain in a contrastive manner.

## 1 Introduction

In order to build reliable and trustworthy NLP applications, it is crucial to make models transparent and explainable. Some use cases require the explanations not only to be *faithful* to the model's inner workings but also *plausible* to humans. We follow the terminology from DeYoung et al. (2020) and define *plausible* explanations as model-based rationales that have high agreement with human rationales, and *faithful* explanations as the input tokens most relied upon for classification. Both qualities (plausibility and faithfulness) can be estimated via metrics, i.e., are not binary. Recently, various contrastive explanation approaches have been proposed in NLP (Jacovi et al., 2021; Paranjape et al.,

---
∗ Equal contribution.

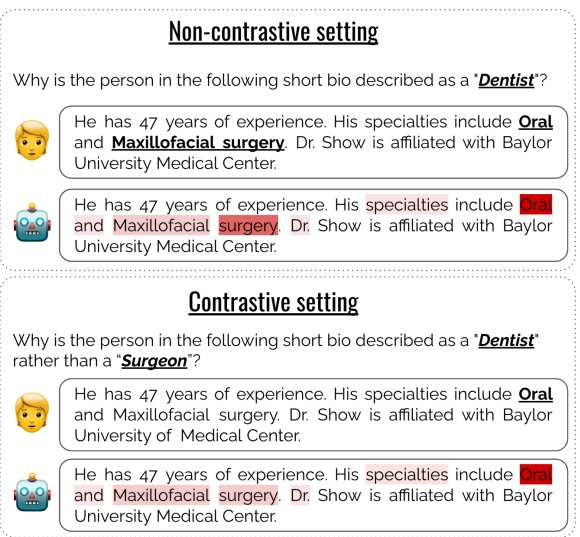

Figure 1: An example from the BIOS dataset of *non-contrastive* and *contrastive* human and model-based rationales. Human rationales are underlined and bold-faced, while model-based rationale attribution scores are highlighted in red (positive) or blue (negative) colors.

2021; Yin and Neubig, 2022) where an explanation for a model's decision is provided *in contrast to* an alternative decision. Figure 1 shows an example for text classification of professions, where the contrastive setting is phrased as: *"Why is the person [...] a dentist rather than a surgeon?"*. Contrastive explanations are considered closer to how humans would argue and thus considered more valuable for humans to understand the model's decision (Lipton, 1990; Miller, 2019; Jacovi et al., 2021). In particular, the latter has been shown in previous evaluation studies. So far however, it has not been empirically investigated whether contrastive explanations are indeed closer to how humans would come to a decision, i.e., whether human rationale annotations are more similar to contrastive explanations than to non-contrastive explanations.

In this work, we initially compare human gaze and human rationales collected from a subset of the SST2 dataset (Socher et al., 2013; Hollenstein

et al., 2018; Thorn Jakobsen et al., 2023), a binary sentiment classification task, with contrastive and non-contrastive model-based explanations using the Layer-wise Relevance Propagation (LRP, Bach et al. 2015; Ali et al. 2022) framework. We find no difference between non-contrastive and contrastive model-based rationales in this binary setting. The analysis on DynaSent (Potts et al., 2021) yields similar results. We thus further explore the potential of contrastive explanations, collecting human rationale annotation for both settings, contrastive and non-contrastive, on a subset of the BIOS dataset (De-Arteaga et al., 2019) for a five-way medical occupation classification task, and compare them to model-based explanations.

We find that human annotations in both settings agree on a similar level with model-based rationales, which suggests that similar tokens are selected. Contrastive human explanations seem to be more specific (fewer words annotated), but agreement between the two settings varies across classes. Based on these results, we conclude that humans do not necessarily explain in a contrastive manner by default. Moreover, model explanations computed in a non-contrastive and contrastive manner do not differ while both align equally well with human rationales. We observe similar findings in another single-label multi-class animal species classification dataset, DBPedia Animals.

**Note – Human rationales ≠ reasoning:** As part of this work, we collect *human rationales* in the form of highlighting supporting evidence in the text to decide for the gold label; we show an example in Figure 1. These human rationales should be understood as proxies for how humans (annotators) explain (rationalize) a given outcome *post-hoc*, which shall not be conflated with how humans reason, came to a decision, *ad-hoc*. In other words, human rationales can be only be seen as a filtered aftermath of human reasoning. Hence, our observations are only suggestive of how humans explain decisions they are provided with, rather than how they come to make these decisions. The latter could possibly be examined by analyzing physiological signals, e.g., brain stimuli or gaze (eye-tracking), *pre-hoc* in relation to rationales.

**Contributions** The main contributions of this work are: (i) We provide an extensive comparison between contrastive and non-contrastive rationales provided both by humans and models for three different model architectures (RoBERTa, GTP2, T5)

and sizes (small, base, large) and for three different post-hoc explanation methods (LRP, GradientxInput, Gradient Norm) on four English text classification datasets. (ii) We include both human annotations and gaze patterns into our analysis, which provide human signals at different processing levels. (iii) We release a subset of the BIOS dataset, a text classification dataset for five medical professions. (iv) We further release human rationale annotations for 100 samples of this newly released dataset for both contrastive and non-contrastive settings.

We release our code on Github to foster reproducibility and ease of use in future research.[1]

## 2 Related Work

**Contrastive Explainable AI (XAI)** Contrastive explanations have only recently been applied in language models. We are revising the most prominent recent papers but also would like to refer to earlier work in the field of computer vision (Dhurandhar et al., 2018; Prabhushankar et al., 2020). Jacovi et al. (2021) propose a framework to generate contrastive explanations by projecting the latent input representation to a maximally contrastive space. They evaluate the usability of contrastive explanation on capturing bias on text classification benchmarks BIOS and NLI, but not the quality of the explanations per se. We use the BIOS dataset to collect human rationales in a contrastive and non-contrastive setting. Paranjape et al. (2021) apply and human-evaluate contrastive explanations in a commonsense-reasoning task. They find contrastive explanations to be more useful to humans and that model performance can be improved via conditioning predictions using contrastive explanations. Yin and Neubig (2022) compare three contrastive explainability methods with their original version: Gradient Norm, InputxGradient and Input Erasure. They apply their methods in different settings including a user study for predicting the language model's behaviour and conclude that contrastive explanations are both more intuitive and fine-grained in comparison to non-contrastive explanations. We use their methods in our analysis together with (a contrastive version of) LRP (Gu et al., 2018) extended to Transformer models.

Both aforementioned human evaluation studies differ to our evaluation analysis as they provide explanations to humans to ask about the model deci-

---

[1]https://github.com/coastalcph/humans-contrastive-xai

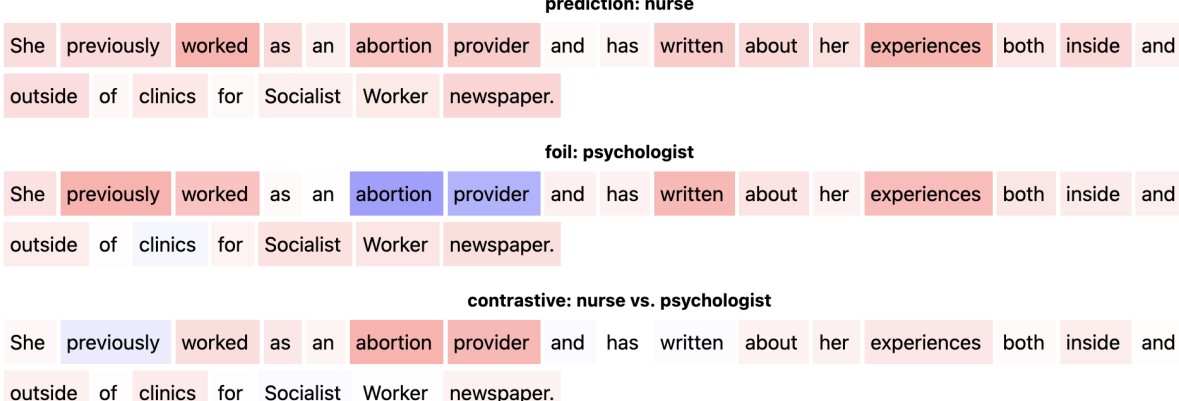

Figure 2: An example biography for a 'nurse' from the BIOS dataset highlighted with LRP relevance scores -red for positive, and blue for negative- per class (occupation) based on RoBERTa large. We further show explanations for the foil ('psychologist'). In the last row, we present an explanation for 'nurse', the correct outcome, in contrast to 'psychologist', the second best guess of the model.

sion and the relevance and helpfulness, respectively. In contrast, we aim to analyze whether the rationales provided by humans are closer to contrastive explanations than non-contrastive explanations.

**Human rationales vs. model rationales** Human rationale annotations often serve as a reference point for evaluating model explanations by directly comparing them to the human ground truth (Schmidt and Bießmann, 2019; Camburu et al., 2018; DeYoung et al., 2020). While high agreement between model-based and human explanations does not necessarily lead to a faithful model prediction (Rudin, 2019; Atanasova et al., 2020), others argue that providing explanations plausible to humans is crucial when building trustworthy systems (Miller, 2019; Jacovi et al., 2023).

## 3 Methodology

Non-contrastive explanations typically compute the most relevant features for a target class label $L$, e.g., by computing the gradients with respect to the logit (score) of the top-predicted class. Contrastive explanations can in extension be obtained by considering the difference in evidence for the target class $L$ and some foil class ($F \neq L$). To compute contrastive and non-contrastive explanations, we use the following methods:

**Layer-wise Relevance Propagation** Since naive computation of gradients in Transformer models has been shown to result in less faithful explanations (Ali et al., 2022), we build contrastive explanations in the framework of 'Layer-wise Relevance

Propagation' (LRP) for Transformer models. To compute explanations that accurately reflect the model predictions, the handling of specific non-linear model components is needed, which includes the layer normalization and attention head modules, that can be treated via carefully detaching nodes of the computation graph as part of the forward pass. For non-linear activation functions, i.e., GeLU, we propagate relevance proportionally to the observed activations (Eberle, 2022).

**Gradient $\times$ Input** We further compute contrastive and non-contrastive explanations via 'Gradient $\times$ Input' (Baehrens et al., 2010; Shrikumar et al., 2017), which can be seen as a special case of LRP without the use of specific propagation rules. In our setting this results to no specific treatment of non-linear computations, i.e., detaching of non-conserving modules.

**Gradient Norm** In addition, computing the norm of the gradient (Li et al., 2016) directly has been also considered in the context of contrastive explanations by Yin and Neubig (2022).

To obtain contrastive explanations, we define the evidence to be explained as the difference:

$$y(x)_l - y(x)_f,$$

where $y(x)_l$ is the logit (score) of the top-predicted target label by the model and $y(x)_f$ is the score for the foil. For generative models, we select the logit of the predicted label, or foil, token.

## 4 Experiments

### 4.1 Datasets

In this study, we conduct experiments on four single-label English classification datasets: SST2 (Socher et al., 2013), DynaSent (Potts et al., 2021), BIOS (De-Arteaga et al., 2019), and DBPedia Animals (Lehmann et al., 2015).

**SST2** The dataset contains approximately 70,000 (68k train/1k dev/1k test) English movie reviews, each labeled with *positive* or *negative* sentiment. To analyze non-contrastive vs. contrastive explanations, we use the non-contrastive human rationale annotations provided by Thorn Jakobsen et al. (2023).[2] The 263 samples chosen by Thorn Jakobsen et al. belong to the development or test split from SST2, and they overlap with the samples from ZuCo, an eye-tracking dataset where reading patterns have been recorded from English native speakers reading movie reviews from SST (Hollenstein et al., 2018).

**DynaSent** This English sentiment analysis dataset contains approximately 122,000 sentences, each labeled as *positive*, *neutral*, or *negative*. Thorn Jakobsen et al. (2023)[2] released non-contrastive annotations for 473 samples from the test set, excluding examples labeled as neutral on the premise that neutral sentiment comes in lack of context, i.e., no evidence of positive or negative sentiment which we use to compare non-contrastive and contrastive model rationales.

**BIOS** The dataset comprises English biographies labeled with occupations and binary genders. This is an occupation classification task, where bias with respect to gender can be studied. We consider a subset of 10,000 biographies (8k train/1k dev/1k test) targeting 5 medical occupations (*psychologist*, *surgeon*, *nurse*, *dentist*, *physician*).

We collect and release human rationale annotations for a subset of 100 biographies in two different settings: non-contrastive and contrastive. In the former, the annotators were asked to find the rationale for the question "*Why is the person in the following short bio described as a L?*", where $L$ is the gold label occupation, e.g., nurse. In the latter, the question was "*Why is the person in the following short bio described as a L rather than a F?*", where $F$ (foil) is another medical occupation, e.g.,

physician. Figure 1 depicts a specific example in both settings. We collect annotations via Prolific,[3] a crowd-sourcing platform. We select annotators with fluency in English and include a pre-selection annotation phase for the contrastive setting, where clear guidelines were provided. We use Prodigy,[4] as the annotation platform, and we change partly the guidelines and the framing of the questions, as shown above, between the two (contrastive and non-contrastive) settings. For each example, we have word-level annotations from 3 individuals (annotators). For further details on the annotation process and the dataset, see Appendix A. We release the new version of BIOS, dubbed *Medical BIOS*, annotated with human rationales on HuggingFace Datasets (Lhoest et al., 2021).[5]

**DBPedia Animals** We consider a subset of the DBPpedia dataset comprising 10,000 (8k train/1k dev/1k test) English Wikipedia article abstracts for animal species labeled with the respective biological class out of 8 classes (*amphibian*, *arachnid*, *bird*, *crustacean*, *fish*, *insect*, *mollusca*, & *reptile*).[6]

### 4.2 Examined models

We consider three publicly available pre-trained language models (PLMs) covering three different architectures: (i) encoder/(ii) decoder-only, and (iii) encoder-decoder. We use RoBERTa of Liu et al. (2019), GPT-2 of Radford et al. (2019), and T5 of Raffel et al. (2020) in three different sizes (small, base, and large); we thus test 9 models in total.[7] We fine-tune RoBERTa and GPT-2 using a standard classification head, while we train T5 with teacher-forcing (Williams and Zipser, 1989) as a sequence-to-sequence model. We conduct a grid search to select the optimal learning rate based on the validation performance. We use the AdamW optimizer for RoBERTa, and GPT-2 models, and Adafactor for T5, following Raffel et al. (2020). We use a batch size of 32 examples and train our classifiers up to 30 epochs using early stopping based on validation performance.

---

[2]https://huggingface.co/datasets/coastalcph/fair-rationales

[3]https://www.prolific.co/
[4]https://prodi.gy/
[5]https://huggingface.co/datasets/coastalcph/medical-bios
[6]https://huggingface.co/datasets/coastalcph/dbpedia-datasets
[7]We report the classification performance and the number of parameters per model in Table 2.

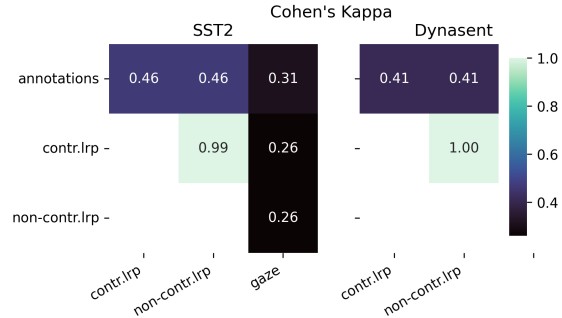

Figure 3: Comparison between model rationales with human annotations for binary sentiment classification on DynaSent and additionally with human gaze on SST2.

### 4.3 Aggregating rationales

For all examined datasets (Section 4.1), we consider the following aggregation methodology (see Figure 9 in App. A) for human and model rationales, before we proceed with the analysis:

- Human annotations are aggregated based on a word-based majority vote across all annotators.

- Model explanations, computed at the sub-word level via an XAI method, are aggregated per word via max-pooling (Eberle et al., 2022).

- When comparing with human rationales, model rationales and gaze are binarized based on the top-k scored tokens, where $k$ is the number of tokens selected in the aggregated human rationales.

## 5 Results

We first show results for the binary sentiment classification datasets, SST2 and DynaSent, before we dive into the extensive analysis of the human and model-based rationales of the BIOS dataset.

### 5.1 Sentiment classification tasks

We show results for SST2 and DynaSent in Figure 3 in the form of agreement scores computed with Cohen's Kappa for gaze, human annotations and the contrastive and non-contrastive LRP scores for RoBERTa-base. For SST2, we find that gaze shows higher agreement with human annotations than with model rationales (0.31 vs. 0.26) whereas the agreement between annotation and model rationales is even higher (0.46). We see a very high agreement (0.99) between contrastive and non-contrastive model rationales. The analysis for DynaSent shows similar results with a lower agreement between annotations and model rationales (0.41). The numbers show an almost perfect agreement on the binarized versions of the

model rationales between the contrastive and the non-contrastive settings but we also see a correlation $> 0.99$ for the continuous values for both datasets. The reason for this might be that in binary classification settings, LRP already considers the only alternative when assigning importance scores, i.e., already computes evidence for one class in contrast to the only other class. For the rest of the paper, we therefore focus on the other two datasets which include 5 and 8 classes, respectively.

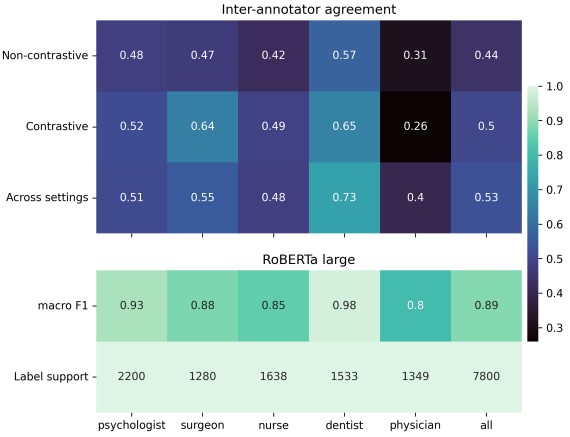

Figure 4: *Upper:* Cohen's Kappa scores for inter-annotator agreement for human rationale annotation (i) averaged across pairwise comparison within non-contrastive setting, (ii) averaged across pairwise comparison within contrastive setting, (iii) between contrastive and non-contrastive setting. *Lower:* Model performance scores (macro F1) for the best model (RoBERTa-large) and training support (#samples) across BIOS classes.

### 5.2 Human rationales

Initially, we perform an analysis of the collected human rationales on the BIOS data by comparing the two settings (contrastive and non-contrastive). On average the contrastive rationales are shorter (4 vs. 8 annotated words), which is an indicator of more precise (focused) rationales. This is expected, since the annotators in the contrastive setting were asked to explain the decision for one class in contrast to another, e.g., 'surgeon' against 'physician'. For instance, in the following example (biography) describing a 'surgeon':

> *"After earning his medical degree, virtually all his training has been concentrated on two fields: **facial plastic and reconstructive surgery**, and **head and neck surgery** — otherwise known as Otolaryngology."*

the terms 'medical degree' and 'Otolaryngology'

were both annotated in the non-contrastive annotation setting (marked in underline), but not in the contrastive one (marked in **bold**).

In Figure 4, we present the inter-annotator agreement measured by Cohen's Kappa within but also between contrastive and non-contrastive human rationales. We observe similar results for all three scores per class with scores ranging from 0.4 to 0.73 for the comparison across contrastive and non-contrastive settings. The class *physician* shows lowest scores in all three comparisons whereas *dentist* achieves the highest agreement in all 3 comparisons. This indicates that the agreement across settings is similar to the agreement within settings and thus the selection of tokens does not necessarily differ between contrastive and non-contrastive annotations.

The low agreement score for the class *physician* can be explained by the lack of keywords in the biographies, similar to *nurse* as those professions do not necessarily imply a medical specialization and vary also across countries. The biographies with the label *dentist* and *surgeon* often include very clear keywords, some even semantically related to the profession, which makes identifying them much easier for humans, as well as for models.[8] In the lower part of Figure 4, we also show label support in the train data and macroF1-scores for the best-performing model, RoBERTa-large. The F1-scores show a similar distribution across classes where *dentist* almost reaches perfect accuracy with a macroF1-score of $0.98$ and *physician* with the lowest score of $0.8$. In other words, both humans and models face similar challenges.

### 5.3 Human vs. model-based rationales

We further proceed with an analysis of model-based rationales compared to human rationales. In Figure 5, we present the agreement between human rationales and model-based rationales computed with LRP for the base version of the three different examined models (RoBERTa, GPT-2, and T5). Since LRP provides continuous attribution scores for all tokens, in order to compare with binary human rationales, we binarize the model-

---

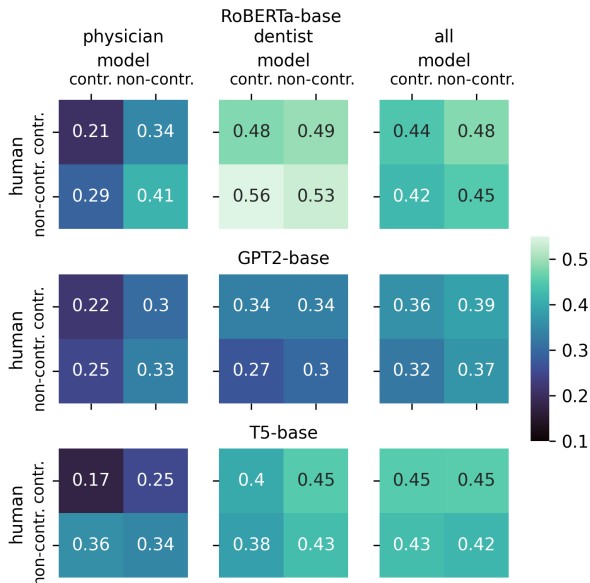

Figure 5: Agreement between human rationales and model-based explanations computed with LRP. Upper: RoBERTa, Center: GPT-2, Lower: T5

based rationales based on the top-k attributed tokens, where k is the total number of selected tokens in the corresponding human rationale. Considering the agreement across all classes, i.e., *all* in Figure 5, we observe that in most cases contrastive and non-contrastive model-based rationales have a similar agreement rate with both contrastive and non-contrastive human rationales (maximum difference is 0.06). In other words, although fewer words are selected by humans in the contrastive setting, the selection of tokens does not seem to be heavily influenced by the two different settings for both models and humans. The agreement is substantially lower in the class 'physician', where highly indicative words are not present, as noted earlier. Here, we also see an overall higher agreement between *non-contrastive* model rationales and human rationales (right column of left-most plots). The original claim, that human rationales are more similar to contrastive than to non-contrastive model rationales (left column vs. right column of all subplots) is not visible in Figure 5.

**POS analysis** To better understand the grammatical structure of human and model-based rationales, we analyze the part-of-speech (POS) tags[9] of rationales in the BIOS data. While human and model-based rationales are mainly formed by nouns and adjectives, models tend to give more importance to verbs compared to annotators, who barely selected

---

[8]Inspecting the human annotations, we observe that specialized words, such as 'dental', and 'surgery' are present and selected across all (100%) examples for dentists, and surgeons, respectively. Contrary for physicians, the generic words 'medical', and 'medicine' are present in 50% of the relevant examples, and selected in 60-70% of those. For nurses, the word 'nursing' is present only in 59% of the relevant examples, and has been selected in 100% of those.

[9]POS tagging is done with spaCy (Honnibal et al., 2020).

| Model Size | RoBERTa | GPT-2 | T5 |
|---|---|---|---|
| BIOS | | | |
| Small (S) | 0.90 | 0.76 | 0.38 |
| Base (M) | 0.88 | 0.74 | 0.48 |
| Large (L) | 0.93 | 0.78 | 0.62 |
| DBpedia-Animals | | | |
| Small (S) | 0.97 | 0.72 | 0.68 |
| Base (M) | 0.99 | 0.70 | 0.79 |
| Large (L) | 0.99 | 0.86 | 0.54 |

Table 1: Spearman correlation between contrastive and non-contrastive model-based rationales on the BIOS (upper part) and DBpedia-Animals (lower part) datasets across all examined models.

them as keywords in their explanations. This behavior is consistent across explainability methods, and both contrastive and non-contrastive explanations.

## 5.4 Model-based rationales

In Table 1, we present the Spearman correlation coefficients between contrastive and non-contrastive model-based explanations across all models for the full test set of the BIOS and the DBPedia-animals datasets. We observe that overall explanations highly correlate, in particular for RoBERTa but also for GPT-2. Large models correlate higher for both datasets, except for DBPedia-animals in T5. This finding suggests that contrastive and non-contrastive model-based explanations do not differ per se in the distribution of importance score. We will further look into the selection of tokens and the sparsity of the model explanation.

**Does gender matter?** We extract the top-5 tokens with the highest importance scores attributed by respective explainability methods for each sample and analyze the amount of gendered words on these tokens.[10] With this, we want to quantify what role gender information plays in the model explanations. While human-based rationales do not contain words with grammatical gender, we find that models *do* rely on these tokens when computing explanations. We examine the relative frequency of words –after aggregating the output tokens (see Section 4.3)– related to 'Male' or 'Female'. Heatmaps in Figure 6 show results for the base versions of all 3 models and all 3 explainability methods for the

---
[10]The gender analysis is based on a publicly available lexicon of gendered words in English. https://github.com/ecmonsen/gendered_words.

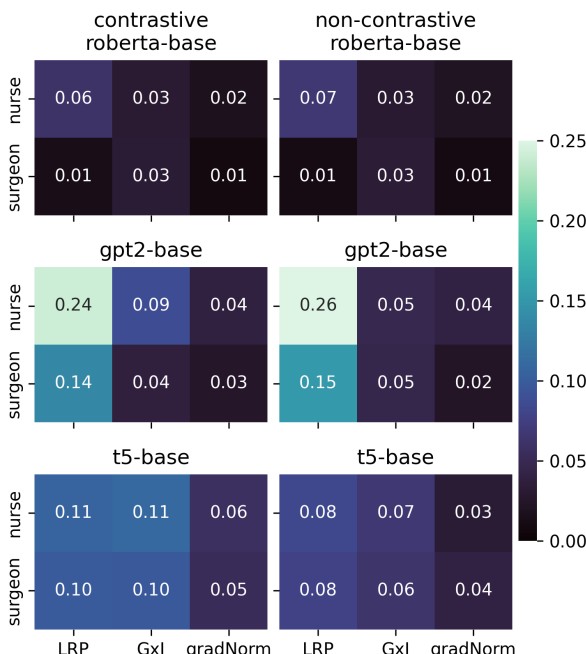

Figure 6: Relative frequency of gendered words among the top-5 tokens in explanations.

two classes with the highest frequency of gendered words in the explanations. Note that these classes, 'nurse' and 'surgeon', also have the highest gender disparity in the dataset (85% male surgeons and 91% female surgeons). We overall see more gendered words in GPT-2 compared to the other models, particularly for explanations computed with LRP. The high dependency on gendered tokens for GPT-2 might be one of the reasons behind the overall lower agreement with human rationales displayed in Figure 5 compared to other models.

**Degree of information in explanations** To better understand the differences between contrastive and non-contrastive explanations in both humans and models, we compute entropy to assess sparsity of the respective attributions. Averaged sentence level entropy values on human rationales reveal that non-contrastive explanations are less sparse, i.e., their averaged entropy is significantly higher (1.72 *vs* 1.14, $p<0.05$). This results in sparser human rationales for contrastive explanations indicating that humans do indeed choose relevant tokens more selectively. When looking at model explanations on the BIOS dataset, we observe different entropy levels across explainability methods, but less so between contrastive and non-contrastive explanations, with the exception of T5 models. These observations provide additional support for findings in text generation that have reported benefits of contrastive

explanations to provide more informative explanations for the prediction of the next token compared to non-contrastive methods (Yin and Neubig, 2022). This would also explain lower correlation coefficients for T5 in Table 1 between contrastive and non-contrastive model explanations.

**Where explanations differ.** We show a hand-picked example of a contrastive model explanation (*nurse* vs. *surgeon*) from RoBERTa large in comparison to the non-contrastive explanation and the foil explanation in Figure 2. Here, we clearly see, that (i) the order of the most important tokens changes from the non-contrastive (upper) to the contrastive (lower) explanation. For instance, the word *abortion* gets the highest score in the contrastive explanation whereas the word *worked* is considered most important in the non-contrastive explanation. We also see a more sparse distribution of the importance scores in the contrastive explanation than in the non-contrastive explanation. We further look into a possible link between model uncertainty, i.e., how close are the class probabilities between the first two classes, and difference in contrastive and non-contrastive explanations, i.e., Spearman correlation coefficient. We find that the two variables highly correlate with each other, in particular for RoBERTa where coefficients range from $0.6 - 0.73$ for LRP. This means, that contrastive and non-contrastive model explanations are more similar when the model is more certain about the label prediction.

## 6 Discussion and Conclusion

In this work, we have compared both human and model rationales in contrastive and non-contrastive settings on four English text classification datasets.

We find that human rationale annotations agree on a similar level within than across contrastive and non-contrastive settings but fewer tokens are selected in the contrastive settings (on average 4 vs. 8). This suggests that there is not per se a difference in token selection for the two settings but tokens are selected more carefully in the contrastive setting. The agreement varies across classes, indicating that for more challenging labels the token selection is not as straightforward as for classes that share a specific vocabulary.

We further compare human rationales with model-based explanations and find no difference in agreement between the contrastive and non-contrastive setting for both models and humans

on the BIOS dataset.

On the binary sentiment classification tasks, we see similar agreement scores between non-contrastive human rationales and model explanations than for the 5-class classification task on BIOS. The numbers need to be compared carefully as the annotation task was different across the two datasets. For the sentiment classification task, no labels were given a-priori and we only analyze the samples where the true label agrees with the label assigned by the annotator. Furthermore, the sentiment classification task is much more subjective than the occupation classification task. Annotators might select tokens differently when they first had to assign a label, i.e., first assess the sentence before deciding to which class it most likely belongs. Including human gaze into the analysis shows lower agreement with the model explanations in comparison to the human annotations. Prior work has shown that human gaze correlates to a higher degree with attention mechanisms (Eberle et al., 2022). In general, human gaze could be considered an alternative to human rationales when evaluating model explanations as they provide more information and the task, i.e., reading the text, might be more intuitive than assigning rationales afterwards.

When comparing model-based explanations with each other, we find them to highly correlate between contrastive and non-contrastive settings. In general, our results did not show that contrastive explanations are by default more class-specific in selecting relevant tokens than non-contrastive explanations. Our analysis suggests that contrastive explanations are more class-specific, i.e., focus on specifc terms for classes that share a joint set of features (similar tokens) like *dentist* and *surgeon* in the BIOS dataset, similar to human rationales. In line with previous work, we have seen that non-contrastive explanations are not necessarily class discriminative and that contrastive explanations can be more class specific but overall share similar features for similar classes (Gu et al., 2018). While we have observed a strong correlation between model-based contrastive and non-contrastive explanations, a qualitative analysis of text samples has in parallel provided sensible examples where contrastive explanations do provide more class-specific information that deviates from the relevant features selected by non-contrastive methods.

Our findings suggest that contrastive explana-

tions are in particular useful in generative models. In contrast to the limited differences observed in the context of few-label classification settings typically investigated, our study provides additional evidence supporting the benefit of the contrastive setting for more complex tasks. This is further supported by findings in self-explaining language models where contrastive prompting leads to explanations that are preferred by humans over non-contrastive explanations (Paranjape et al., 2021).

The subtle differences between contrastive and non-contrastive explanations may provide important signal for improving ML models. Previous work has shown how non-contrastive explanations provide useful information for debugging and removing undesired model behavior (Anders et al., 2022). In extension, contrastive and non-contrastive explanations could be useful during training to improve robustness of models and avoid shortcut learning behavior by regularizing the model to focus on more class-specific features.

## Limitations

Our analysis is limited to English text classification datasets. In order to make more general claims about contrastive explanations, an extension of our analysis to more languages and downstream tasks is needed. The tasks and datasets examined are further limited to a small number of classes, nonetheless not binary as in prior studies, which may affect the efficiency and inherent need for contrastive explanations, since the degree of differentiation between the classes may be too broad, e.g., a dentist and psychologist are two very different medical professions. Experimenting with datasets including hundreds of labels (Chalkidis and Søgaard, 2022; Kementchedjhieva and Chalkidis, 2023), which in many cases are very close semantically, could potentially lead to different results.

Furthermore, we compare model explanations and human rationales both in a *post-hoc* way where first a decision has been made and evidence has been collected afterwards. This is briefly discussed in the introduction. We use a limited definition of plausible explanations, i.e., we compare binary human rationale annotations with continuous model explanations which is not trivial and we automatically filter out information when binarizing the model explanations. For a complementary evaluation, we would also need to show the collected and computed rationales again to human annotators

to further evaluate their plausibility and usability, i.e., are they useful for humans to understand the models, see Brandl et al. (2022); Yin and Neubig (2022).

## Ethics Statement

**Broader Impact.** We release the first dataset with human annotations in both contrastive and non-contrastive settings in NLP. We hope this incentivizes other researchers to further look into contrastive XAI methods and to extend this dataset by other languages and tasks.

**Annotation Process.** All participants were informed about the goal of the study and the procedure at the beginning of the study. They all gave written consent to collect their annotations and demographic data. Participants were paid an average of 12 GBP/hour. The dataset is publicly available.[5] All answers in the study are anonymized and cannot be used to identify any individual.

**Potential risk.** We do not anticipate any risks in participating in this study. We are aware of potential poor working conditions among crowd-workers[11] and try to counteract by paying an above-average compensation.

## Acknowledgements

We thank our colleagues at the CoAStaL NLP group for fruitful discussions in the beginning of the project. In particular, we would like to thank Jonas Lotz, Qinghua Zhao and Yova Kementchedjhieva for valuable comments on the manuscript. OE received funding by the German Ministry for Education and Research (under refs 01IS18056A and 01IS18025A) and BBDC/BZML and BIFOLD. LC, and IC are funded by the Novo Nordisk Foundation (grant NNF 20SA0066568). SB is funded by the European Union under the Grant Agreement no. 10106555, FairER. Views and opinions expressed are those of the author(s) only and do not necessarily reflect those of the European Union or European Research Executive Agency (REA). Neither the European Union nor REA can be held responsible for them.

---

[11]https://www.noemamag.com/the-exploited-labor-behind-artificial-intelligence

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

## A  BIOS Annotations

We collect and release human rationale annotations for a subset of 100 biographies in two different settings: non-contrastive and contrastive. In the non-contrastive setting, the annotators were asked to find the rationale for the question "Why is the person in the following short bio described as a $L$?" where $L$ is the gold label occupation, e.g., nurse. In the contrastive setting, the question was "Why is the person in the following short bio described as a $L$ rather than a $F$?" where $F$ (foil) is another medical occupation, e.g., physician. Figure 1 provides a specific example in both settings.

We used Prodigy,[12] as the annotation platform (Figures 7-8) and made some adjustments to the guidelines and the phrasing of the questions, as shown above, between the two (contrastive and non-contrastive) settings.

We collect annotations via Prolific,[13] an online crowd-sourcing platform. We compensated all annotators at an hourly rate of 12£. We select annotators with fluency in English through a pre-selection annotation phase, where clear guidelines were provided. We provided introductory information and guidelines via Google Forms. The guidelines are the following:

### Guidelines for non-constrastive rationales

- You are going to annotate 30-35 short biographies from people working in the medical sector. The people described in these documents have one of the following medical occupations: 'Psychologist', 'Nurse', 'Physician', 'Surgeon', or 'Dentist'. Each example is paired with a question.

- If you don't feel confident about one of the aforementioned medical occupations, please advice an online open dictionary, such as the Cambridge English Dictionary, and review the definition and some example sentences, e.g., for surgeon: (https://dictionary.cambridge.org/dictionary/english/surgeon).

- See the following question + biography pairs (Test Examples in Figure 7) as examples.

- In the first example, the document (bio) describes them as a 'Dentist'.

[12]https://prodi.gy/
[13]https://www.prolific.co/

- Your task is to find and annotate the words in the bio that answer the following question "Why is the person in the following short bio described as a Dentist?". In other words, which words can be used a evidence that this person is a "Dentist".

- You should select words or phrases (multi-word expressions) that answer this specific question. In other words, your selection should be *valid*, i.e., the words should be related to the given medical occupation and not generic ones.

- You should select ALL the words, or phrases (multi-word expressions) that answer this question. In other words, your annotation should be *complete*, and no words that are evidence of the described medical occupation should be left unannotated.

### Guidelines for constrastive rationales

- You are going to annotate 30-35 short biographies from people working in the medical sector. The people described in these documents have one of the following medical occupations: 'Psychologist', 'Nurse', 'Physician', 'Surgeon', or 'Dentist'. Each example is paired with a question.

- If you don't feel confident about one of the aforementioned medical occupations, please advice an online open dictionary, such as the Cambridge English Dictionary, and review the definition and some example sentences, e.g., for surgeon: (https://dictionary.cambridge.org/dictionary/english/surgeon).

- See the following question + biography pair (Test Examples in Figure 8) as an example.

- In the first example, the document (bio) describes them as a 'Surgeon' rather than a 'Dentist'.

- Your task is to find and annotate the words in the bio that answer the following question "Why is the person in the following short bio described as a Surgeon rather than a Dentist?". Imagine you are trying to convince someone and have to find evidence that this person is a Surgeon and NOT a Dentist (even if in reality both are true).

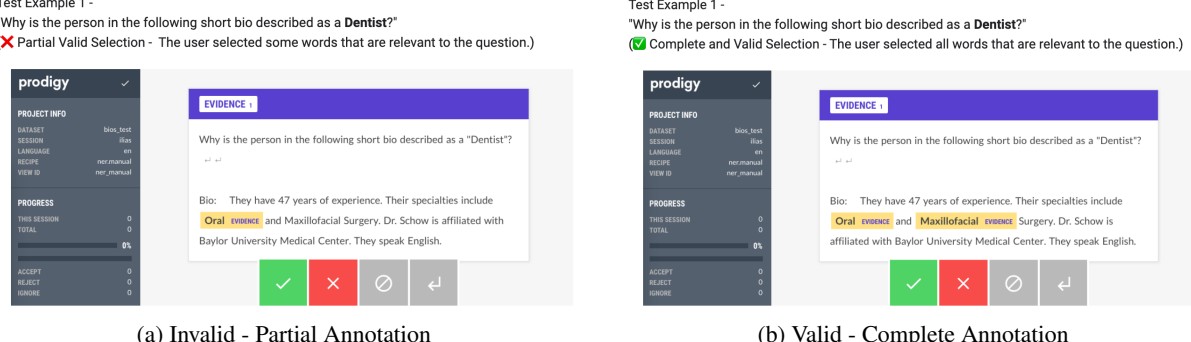

(a) Invalid - Partial Annotation         (b) Valid - Complete Annotation

Figure 7: Example 1 presented as part of the guidelines for the non-contrastive setting.

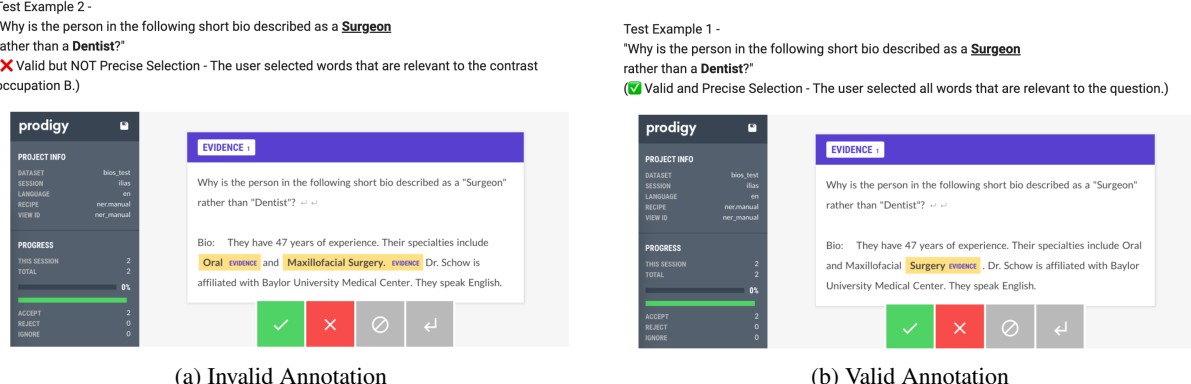

(a) Invalid Annotation         (b) Valid Annotation

Figure 8: Example 2 presented as part of the guidelines for the contrastive setting.

- You should select ALL the words, or phrases (multi-word expressions) that answer this question. In other words, your annotation should be complete, and no words that are evidence of the described medical occupation should be left unannotated.

- You should select ONLY words or phrases (multi-word expressions) that answer why this person is occupation A, e.g., "Surgeon", and not any words that answer the contrast occupation B, e.g., "Dentist". In other words, your selection should be precise, i.e., the words should be related to the given medical occupation and not the one in contrast.

For the contrastive setting, which we believe is more difficult to understand at first, we included a pre-selection process. We therefore conducted a pilot annotation project for 5 straightforward examples. We selected the annotators based on two criteria: (a) manual inspection of their annotations to assess, if they follow the guidelines, (b) computing pair-wise inter-annotator agreement and excluding annotators with low scores (<0.5 Cohen's Kappa).

The selected annotators annotate a final subset

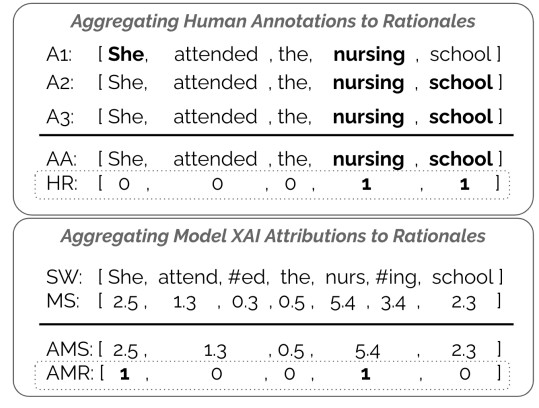

Figure 9: Depiction of aggregation methodology for human and model rationales. Notation: $A_i$ for the $i$th annotator, $AA$ for the aggregated annotation (rationale), $SW$ for sub-words, $MS$ for model XAI attribution scores, $AMS$ for word-level aggregated $MS$, and $AMR$ for aggregated model rationale based on top-k words.

of approx. 35 examples each, in the contrastive setting, approx. 100 annotated examples in total. Overall, we have word-level annotations from 3 individuals (annotators) for each example per setting.

**Aggregating Rationales** In Figure 9, we present an example of the aggregation methodology for human and model rationales.

| Model | | | | Classification Performance | | | |
| Family | Size | Alias | #Params | SST2 | DynaSent | BIOS | DBPedia-Animals |
|---|---|---|---|---|---|---|---|
| RoBERTa | S | MiniLMv2-L6xH768 | 30M | N/A | N/A | 0.872 | 0.976 |
| | M | roberta-base | 125M | 0.929 | 0.879 | 0.880 | 0.982 |
| | L | roberta-large | 355M | N/A | N/A | 0.892 | 0.988 |
| GPT-2 | S | distil-gpt2 | 82M | | | 0.867 | 0.983 |
| | M | gpt2 | 124M | N/A | N/A | 0.869 | 0.983 |
| | L | gpt2-M | 355M | | | 0.881 | 0.992 |
| T5 | S | t5-v1_1-small | 61M | | | 0.886 | 0.985 |
| | M | t5-v1_1-base | 223M | N/A | N/A | 0.897 | 0.989 |
| | L | t5-v1_1-large | 750M | | | 0.887 | 0.989 |

Table 2: Test Results (Micro-F1) for all models (RoBERTa, GPT-2, T5) and all sizes (Small, Base, Large) across all datasets. We also report the number of parameters per model (#Params). Best scores for each model per dataset are underlined.

# B  Additional Results

In Table 2, we report the classification performance for all examined models across all datasets, alongside other model details.