# OpenReview forum: "Rather a Nurse than a Physician - Contrastive Explanations under Investigation"
_EMNLP/2023/Conference — EMNLP 2023 Main_

### Official Review · Reviewer_i43o · 2023-08-03

**Soundness:** 3

**Excitement:**

4: Strong: This paper deepens the understanding of some phenomenon or lowers the barriers to an existing research direction.

**Paper Topic And Main Contributions:**

This paper is about contrastive explanations in NLP models, and it makes several contributions towards understanding how humans and models explain decisions. The paper addresses the problem of making NLP models more transparent and explainable, which is crucial for building reliable and trustworthy NLP applications.

The main contributions of the paper include:

1. Analyzing four English text classification datasets and fine-tuning three different models (RoBERTa, GTP-2, and T5) in three different sizes, and applying three post-hoc explainability methods (LRP, GradientxInput, GradNorm).
2. Collecting and releasing human rationale annotations for 100 samples from the BIOS dataset for contrastive and non-contrastive settings.
3. Conducting a cross-comparison between model-based rationales and human annotations, both in contrastive and non-contrastive settings, and finding a high agreement between the two settings for models as well as for humans.
4. Showing that humans do not necessarily explain in a contrastive manner, and that model explanations computed in both settings align equally well with human rationales.


**Questions For The Authors:**

1. Can you provide more details on the criteria used to select the three models and three post-hoc explainability methods used in the study?
2. Have you considered using other models or methods to evaluate the quality of model explanations? If so, why were they not included in the study?
3. Can you provide a more detailed analysis of the differences between contrastive and non-contrastive explanations, and how they impact the quality of model explanations?
4. How do you define "plausible explanations," and how did you ensure consistency in the evaluation of model explanations?
5. Have you considered the impact of different factors (e.g., dataset size, model size, explainability method) on the quality of model explanations? If so, can you provide a more detailed analysis of these factors?
6. How do you plan to extend this work to other languages and downstream tasks, and what challenges do you anticipate in doing so?
7. Can you provide a more detailed analysis of the collected human rationale annotations, and how they compare to model explanations?
8. How do you plan to make the collected human rationale annotations available to the broader research community, and what potential applications do you see for this dataset?

**Reasons To Accept:**

The strengths of this paper include its thorough analysis of contrastive explanations in NLP models, its use of multiple datasets and models, and its collection of human rationale annotations for contrastive and non-contrastive settings. The paper also provides valuable insights into how humans and models explain decisions, and highlights the importance of making NLP models more transparent and explainable.

If this paper were to be presented at the conference or accepted into Findings, the main benefits to the NLP community would be:

1. Providing a better understanding of how humans and models explain decisions, which can inform the development of more transparent and trustworthy NLP applications.
2. Releasing a new dataset with human annotations in both contrastive and non-contrastive settings, which can incentivize other researchers to further look into contrastive XAI methods and to extend this dataset by other languages and tasks.
3. Demonstrating the effectiveness of multiple explainability methods and model sizes, which can help guide future research in this area.
4. Highlighting the importance of collecting human rationale annotations for evaluating model explanations, which can improve the interpretability and accountability of NLP models.

**Reasons To Reject:**

1. The study only uses three models and three post-hoc explainability methods, which may not be representative of the full range of models and methods used in NLP research.
2. The study does not provide a detailed analysis of the differences between contrastive and non-contrastive explanations, which may limit the usefulness of the findings for practitioners.
3. The study does not provide a clear definition of what constitutes a "plausible explanation," which may lead to inconsistencies in the evaluation of model explanations.
4. The study does not provide a detailed analysis of the impact of different factors (e.g., dataset size, model size, explainability method) on the quality of model explanations, which may limit the usefulness of the findings for future research.
5. The release of the labeled subset seems to be too small, i.e., 100. Can you elaborate it?


**Reproducibility:**

3: Could reproduce the results with some difficulty. The settings of parameters are underspecified or subjectively determined; the training/evaluation data are not widely available.

**Reviewer Confidence:**

3: Pretty sure, but there's a chance I missed something. Although I have a good feel for this area in general, I did not carefully check the paper's details, e.g., the math, experimental design, or novelty.

---

> ### Author Rebuttal · Authors · 2023-08-28
>
> Thank you for your review, we address your questions below, which also address/cover your comments.
>
> > 1. Can you provide more details on the criteria used to select the three models and three post-hoc explainability methods used in the study?
>
> We selected these models to cover all widely used transformer-based model architectures (encoder-only, decoder-only, encoder-decoder) in 3 different sizes. We favored variety in these dimensions over different pre-trained models, e.g., BERT and RoBERTa, in similar settings (architecture, size).
>
> > 2. Have you considered using other models or methods to evaluate the quality of model explanations? If so, why were they not included in the study?
>
> As mentioned above, we cover a wide range of model architectures, in different model sizes, and many post-hoc XAI methods. We believe that our current metric to evaluate explanations is in line with the literature but are happy to extend this if you have any suggestions.
>
> > 3. Can you provide a more detailed analysis of the differences between contrastive and non-contrastive explanations, and how they impact the quality of model explanations?
>
> This depends on how one defines the quality of model explanations. In this paper, we are interested in the plausibility of explanations, i.e., how well they align with human rationales. We find that they agree on a similar level and discuss this in Section 6 of our paper in more detail.
>
> > 4. How do you define "plausible explanations," and how did you ensure consistency in the evaluation of model explanations?
>
> We define plausibility as the level of similarity between model explanations and rationales provided by humans and compute the F1 score between the two as has been described in DeYoung et al. (2020). We optimize models under similar settings following a grid search over the main hyper-parameters to guarantee a fair ground for comparison, and we apply all XAI methods in a similar fashion.
>
> > 5. Have you considered the impact of different factors (e.g., dataset size, model size, explainability method) on the quality of model explanations? If so, can you provide a more detailed analysis of these factors?
>
> We looked into the human-model alignment, i.e., plausibility, for different model sizes and could not find any consistency, e.g., larger/smaller size influencing plausibility. This can however be due to additional confounding factors such as the model classification performance, or how XAI methods are affected by being applied in larger models. We believe that a detailed analysis of all those factors exceeds the scope of the paper and could be included in follow-up work.
>
> > 6. How do you plan to extend this work to other languages and downstream tasks, and what challenges do you anticipate in doing so?
>
> As NLP research is increasingly moving towards generative instruction-finetuned models, we believe that this should be the main focus for future work.
>
> > 7. Can you provide a more detailed analysis of the collected human rationale annotations, and how they compare to model explanations?
>
> We provide a detailed description of the rationale annotations in the appendix but are happy to complete it in case something is missing. Figure 3 and Figure 5 show results on the comparison between model rationales and human rationales for all 4 datasets, and we comment on those in the respective sections (5.2 and 5.3).
>
> > 8. How do you plan to make the collected human rationale annotations available to the broader research community, and what potential applications do you see for this dataset?
>
> Annotations as well as the code for our experiments will be made available upon publication, most likely via HuggingFace for ease of use. We believe the collected rationales can be useful for future work in contrastive explanations as such a dataset does not exist yet but also for investigating bias where gendered attributes are relevant.

---

### Official Review · Reviewer_vveg · 2023-08-05

**Soundness:** 3

**Excitement:**

4: Strong: This paper deepens the understanding of some phenomenon or lowers the barriers to an existing research direction.

**Paper Topic And Main Contributions:**

The paper mainly analyzes the difference between contrastive and non-contrastive rationales provided by humans and models, using three different model architectures and three different post-hoc explanation methods, on four text classification datasets. The authors find that human annotations agree on a similar level with model-based rationales, in both contrastive and non-contrastive settings. In addition, contrastive human rationales are more specific with fewer words annotated than non-contrastive ones. For model explanations, the rationale difference between the two settings is relatively small except for generation models like T5.

**Questions For The Authors:**

A.	In Tables 1 and 2, I can see that T5-large has a higher correlation than T5-small on the BIOS dataset, but on the DBpedia-Animals, the situation is quite the opposite. Why does T5 show a different correlation in the model size between the two datasets? Could you somehow give an explanation?

**Reasons To Accept:**

1.	The paper draws attention to the comparison between contrastive and non-contrastive rationales provided by humans and models, which provides empirical support for the exploration of using contrastive settings to improve models.
2.	The analysis was conducted based on three model architectures and three post-hoc explanation methods on four datasets, which shows the generality of conclusions.


**Reasons To Reject:**

1.	In contrastive settings, the degree of difference between positive entities and negative ones might be a quite sensitive factor for models to provide different rationales. However, I cannot find such an analysis in the paper. Apart from the comparison between contrastive and non-contrastive rationales, I would prefer to see more discussions about the rationale difference between simple contrastive settings, e.g., nurse vs. teacher (easy to distinguish), and hard ones, e.g., nurse vs. doctor (hard to distinguish).
2.	For complex tasks, contrastive settings might show benefits and provide more information for models. I am curious about how the model rationale would change when the label prediction changes by shifting non-contrastive settings to contrastive ones. I suggest adding such case studies, where in one case, the model has different label predictions between contrastive and non-contrastive settings, while in another case, the model gives the same label prediction between two settings. Then, analyze the difference of their rationale changes.


**Reproducibility:**

3: Could reproduce the results with some difficulty. The settings of parameters are underspecified or subjectively determined; the training/evaluation data are not widely available.

**Reviewer Confidence:**

3: Pretty sure, but there's a chance I missed something. Although I have a good feel for this area in general, I did not carefully check the paper's details, e.g., the math, experimental design, or novelty.

---

> ### Author Rebuttal · Authors · 2023-08-28
>
> Thank you for your comments, we address them below.
>
> > In contrastive settings, the degree of difference between positive entities and negative ones might be a quite sensitive factor for models to provide different rationales. However, I cannot find such an analysis in the paper. Apart from the comparison between contrastive and non-contrastive rationales, I would prefer to see more discussions about the rationale difference between simple contrastive settings, e.g., nurse vs. teacher (easy to distinguish), and hard ones, e.g., nurse vs. doctor (hard to distinguish).
>
> We agree that contrastive explanations are likely most useful for complex tasks, this motivated us initially to look into medical professions only, instead of all professions originally presented in BIOS. In our analysis, we could see that contrastive and non-contrastive explanations are more similar for confident decisions, i.e., when the probability for the true label is clearly higher than for all other labels which is in line with the reviewer's intuition. We will further highlight this point in the revised version.
>
> > For complex tasks, contrastive settings might show benefits and provide more information for models. I am curious about how the model rationale would change when the label prediction changes by shifting non-contrastive settings to contrastive ones. I suggest adding such case studies, where in one case, the model has different label predictions between contrastive and non-contrastive settings, while in another case, the model gives the same label prediction between two settings. Then, analyze the difference of their rationale changes.
>
> We compute all explanations post-hoc which means they have no effect on the model prediction and do not provide any information to the model. It would indeed be interesting to see how contrastive rationales influence the outcome of self-explainable model architectures that rely on rationales for the model prediction.

---

### Official Review · Reviewer_beUE · 2023-08-08

**Soundness:** 4

**Excitement:**

4: Strong: This paper deepens the understanding of some phenomenon or lowers the barriers to an existing research direction.

**Paper Topic And Main Contributions:**

This paper presents an analytic investigation into the effectiveness and the practical relevance of "contrastive explanations" compared to non-contrastive explanations. While the recent literature has assumed that the contrastive explanations are probably more useful and natural for humans, the empirical findings of this paper reveals that this is not necessarily the case. While the findings and conclusions are not ground-breaking per say, this paper presents a carefully conducted study that other researchers might find insightful.

**Reasons To Accept:**

- This paper presents a carefully conducted study that provides new insights on free-form rationales and explanations. Contrary to the common assumptions in the field that contrastive explanations are preferable over non-contrastive explanations, this paper finds that the benefits of contrastive explanations are rather mixed.

**Reasons To Reject:**

- I don't have any significant reason to list for rejection; While I personally don't find the conclusion of this paper particularly surprising or ground-breaking per say, the scope and the substance of the work presented here seems solid for acceptance.

**Reproducibility:**

4: Could mostly reproduce the results, but there may be some variation because of sample variance or minor variations in their interpretation of the protocol or method.

**Reviewer Confidence:**

4: Quite sure. I tried to check the important points carefully. It's unlikely, though conceivable, that I missed something that should affect my ratings.

---

> ### Author Rebuttal · Authors · 2023-08-28
>
> Thank you for your review and positive assessment of our paper.

---

### Meta-Review · Area_Chair_3aD5 · 2023-09-16

**Recommendation:** 5

**Metareview:**

This paper presents a study comparing human and model rationales in both contrastive and non-contrastive settings. The results show that model-based rationales computed in both settings align equally well with human rationales, contradicting a claim in literature that contrastive explanations are more natural for humans than non-contrastive explanations. Overall, the reviewers were impressed by the carefully conducted study on several datasets, architectures, and post-hoc explanation methods. The insights from the study are informative to the community, whereas the dataset used in this paper will be publicly released as a unified benchmark fostering future research. However, the reviewers posted some questions about the justification for the conducted experimental settings and speculated results for alternative settings which the authors answered in the rebuttal.

---

### Decision · Program_Chairs · 2023-10-07

**Decision:**

Accept-Main

**Comment:**

This paper presents a study comparing human and model rationales in both contrastive and non-contrastive settings. The results show that model-based rationales computed in both settings align equally well with human rationales, contradicting a claim in literature that contrastive explanations are more natural for humans than non-contrastive explanations. Overall, the reviewers were impressed by the carefully conducted study on several datasets, architectures, and post-hoc explanation methods. The insights from the study are informative to the community, whereas the dataset used in this paper will be publicly released as a unified benchmark fostering future research. However, the reviewers posted some questions about the justification for the conducted experimental settings and speculated results for alternative settings which the authors answered in the rebuttal.